# Real-Time NMPC for Speed Planning of Connected Hybrid Electric Vehicles

Fei Ju [1], Yuhua Zong [1], Weichao Zhuang [2], Qun Wang [1] and Liangmo Wang [1,*]

1   School of Mechanical Engineering, Nanjing University of Science and Technology, Nanjing 210094, China
2   School of Mechanical Engineering, Southeast University, Nanjing 211189, China
*   Correspondence: liangmowang_njust@163.com

**Abstract:** Eco-cruising is considered an effective approach for reducing energy consumption of connected vehicles. Most eco-cruising controllers (ECs) do not comply with real-time implementation requirements when a short sampling interval is required. This paper presents a solution to this problem. Model predictive control (MPC) framework was applied to the speed-planning problem for a power-split hybrid electric vehicle (HEV). To overcome the limitations of time-domain MPC (TMPC), a nonlinear space-domain MPC (SMPC) was proposed in the space domain. A real-time iteration (RTI) algorithm was developed to accelerate nonlinear SMPC computations via generating warm initializations and subsequently forming the SMPC-RTI. Proposed speed controllers were evaluated in a hierarchical EC, where a heuristic energy management strategy was selected for powertrain control. Simulation results indicated that the proposed SMPC yields comparable fuel savings to the TMPC and the globally optimal solution. Meanwhile, SMPC reduced MPC computation time by 41% compared to TMPC, and SMPC-RTI further reduced MPC computation time without compromising optimization. During the hardware-in-loop (HIL) test, the mean computation time was 9.86 ms, demonstrating potential for real-time applications.

**Keywords:** hybrid electric vehicle; speed planning; dynamic programming; model predictive control; real-time iteration





## 1. Introduction

Electrification of vehicles is considered the most promising solution for reducing both fossil-fuel consumption and pollutant emissions produced by conventional vehicles [1]. Over the past decade, electric vehicles (EVs) and hybrid electric vehicles (HEVs) have experienced explosive growth, and their penetration of the market is expected to continue to rise [2]. Even though the market is prosperous, fuel and electricity consumption are still enormous, which is not environmentally friendly [3]. Therefore, from an application perspective, improving energy efficiency is crucial to realizing sustainable transportation [4].

An effective way to optimize a vehicle powertrain's efficiency is through energy management (EM). Utilizing current demand power or predicted demand power, EM strategy is capable of regulating each component's output while minimizing energy consumption [5]. Nevertheless, performance of EM is dependent on the driving cycle. EM generally indicates significant energy reductions in the city cycle but poor reductions in the highway cycle [6].

An additional way to improve energy-saving capabilities of electrified vehicles is to implement eco-driving techniques [7]. With the aid of network connectivity, the host vehicle can easily access information about the surrounding environment, allowing it to regulate its speed in an eco-friendly manner [8]. In the current literature, eco-driving is classified into two types: green-light optimal speed advisory (GLOSA) and eco-cruising. GLOSA is designed to reduce stops and abrupt accelerations based on upcoming traffic signals generated by vehicle-to-infrastructure (V2I) technology [9,10]. For eco-cruising, global navigation satellite system (GNSS) information is used to calculate optimal speed on various terrain types [11].

Eco-cruising methods have gained considerable attention in recent years, in part due to their ability to save energy, both for individual vehicles and for convoys [12,13]. Among these methods, dynamic programming (DP) is the most widely used, due to its ability to obtain the global optimum [14]. Because of the heavy computational burden induced by dimensionality, the DP method can only be used for offline evaluation or for benchmarking of other methods. In [15,16], both iterative DP and approximate DP were proposed to reduce computational burden. In spite of this, these DP-based methods are not suitable for real-time implementation. Another popular optimal theory that has been widely used for eco-cruising is Pontryagin's maximum principle (PMP). Using PMP, it is possible to derive analytical solutions on a flat road [17]. When road slope is considered, PMP is required to solve a two-point boundary value problem iteratively, which is time-consuming. With a sampling interval of 0.5 s and a time horizon of 20 s, PMP's average computation time was 0.46 s, which is regarded as clearly too long [18]. Reinforcement learning (RL) has been extensively studied for problems associated with eco-driving [19]. The Bellman equation is used in RL to optimize the cost-to-go value function. Because RL introduces approximations to improve computation efficiency, it can be used as a real-time eco-cruising controller [20]. As it is difficult to design a reward function that balances exploration and exploitation, RL may not provide satisfactory results for optimization [21].

The look-ahead control method, also known as model predictive control (MPC), incorporates both optimization performance and computational efficiency, and is thus regarded as one of the most promising methods in the associated field [22,23]. Problems in eco-cruising are typically formulated as linear time-domain MPC [18]. Since road slope depends solely on the position of the vehicle, it is necessary to estimate road slope during the implementation of time-domain MPC. This is the principal limitation of the time-domain method [24]. Through translation of time-domain MPC to space-domain MPC, this issue can be resolved [25,26]. However, this results in space-domain MPC that is no longer linear but rather nonlinear. An important obstacle to practical application of nonlinear MPC (NMPC) is difficulty of solving the optimal control problem (OCP) in real time [27]. Several customized algorithms have been proposed to respond to this challenge, including DP [28], which is fast, and the alternating direction method of multipliers (ADMM) algorithm [29]. Based on reported computation times, these algorithms are still inapplicable when the system sampling interval is short, such as 0.1 s.

Motivated by the discussion above, we propose a real-time NMPC for speed planning of a connected HEV. The main contributions of this paper are threefold: (1) Speed planning with forthcoming road information was formulated as NMPC in the space domain. NMPC captured accurate system behavior and avoided any approximations of road information that may have caused errors. (2) To accelerate calculation of NMPC, a real-time iteration algorithm is proposed. Through comparison of different speed controllers, efficacy of the RTI algorithm is revealed and analyzed. (3) We also introduced a hierarchical eco-cruising controller (EC) for the power-split HEV to evaluate proposed speed controllers. Simulations as well as hardware-in-loop (HIL) tests were conducted to assess fuel savings and real-time performance.

The remainder of this paper is organized as follows: Section 2 introduces the HEV model as well as the EC, which incorporates the high-level speed controller and the low-level powertrain controller. The MPC for speed planning is formulated in Section 3, where the RTI algorithm is also proposed. Results of simulations on a realistic road, as well as discussion, are presented in Section 4, while HIL tests are presented in Section 5. Section 6 concludes this paper.

## 2. HEV Model and Hierarchical EC

This study focuses on the power-split powertrain, which is the most common HEV powertrain. First, we present longitudinal dynamics of the vehicle, followed by modeling of the power-split powertrain. Next, a hierarchical EC is developed.

### 2.1. Vehicle Longitudinal Dynamics

The forces that retard a vehicle during driving include rolling resistance, gradient resistance, and air drag. Given vehicle acceleration, torque required by the vehicle at the powertrain output shaft can be derived as

$$T_o = \frac{r}{k_f}\left[ma + mg(\sin\alpha + f\cos\alpha) + \frac{1}{2}C_D A_f \rho v^2\right], \tag{1}$$

where $r$, $k_f$, $m$, $g$, and $v$ are wheel radius, final reduction ratio, vehicle mass, gravity acceleration, and vehicle velocity, respectively, and $\alpha$ denotes road slope. The parameters $f$, $C_D$, $A_f$, and $\rho$ denote rolling-friction coefficient, aerodynamic drag coefficient, vehicle frontal area, and air density, respectively.

### 2.2. Power-Split Powertrain

An identifying characteristic of the power-split powertrain is that it employs planetary gear sets (PGSs) to decouple engine speed from vehicle speed. Therefore, the combustion engine can always operate at high efficiency in the power-split powertrain. Figure 1 depicts the lever diagram and power flow of the power-split HEV powertrain: $T$, $\omega$, and $F$ represent torque, rotational speed, and internal gear force of PGSs, respectively. In addition, $R$ and $S$ represent the ring radius and sun gear radius of PGS1 and PGS2, and subscripts $e$, $mg2$, $mg2$, $o$, 1, and 2 identify the engine, motor/generator 1 (MG1), motor/generator 2 (MG2), the powertrain output shaft, PGS1, and PGS2.

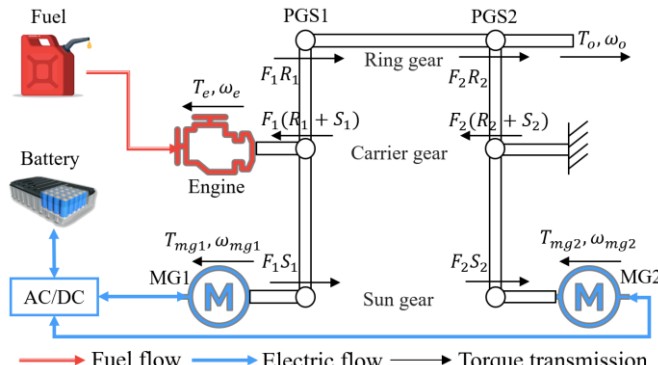

**Figure 1.** Lever diagram and power flow of the power-split HEV powertrain.

It is evident that PGS1 serves as a power-coupling device between the engine and MG1, while PGS2 serves as a reducer for MG2. Since rotational dynamics are much faster than battery dynamics, powertrain losses were ignored in this study [30]. Powertrain dynamics are given by:

$$\omega_o = \frac{vk_f}{r}, \ \omega_{mg2} = \frac{R_2}{S_2}\omega_o \tag{2}$$

$$\omega_{mg1} = \left(\frac{R_1}{S_1} + 1\right)\omega_e - \frac{R_1}{S_1}\omega_o \tag{3}$$

$$T_{mg1} = -\frac{T_e S_1}{R_1 + S_1} \tag{4}$$

$$T_{mg2} = \frac{S_2}{R_2}T_o - \frac{T_e R_1 S_2}{(R_1 + S_1)R_2}. \tag{5}$$

Quasi-static models are usually used to model HEV components. According to the test data in [31], we developed several lookup tables to model powertrain components, as shown in Figure 2.

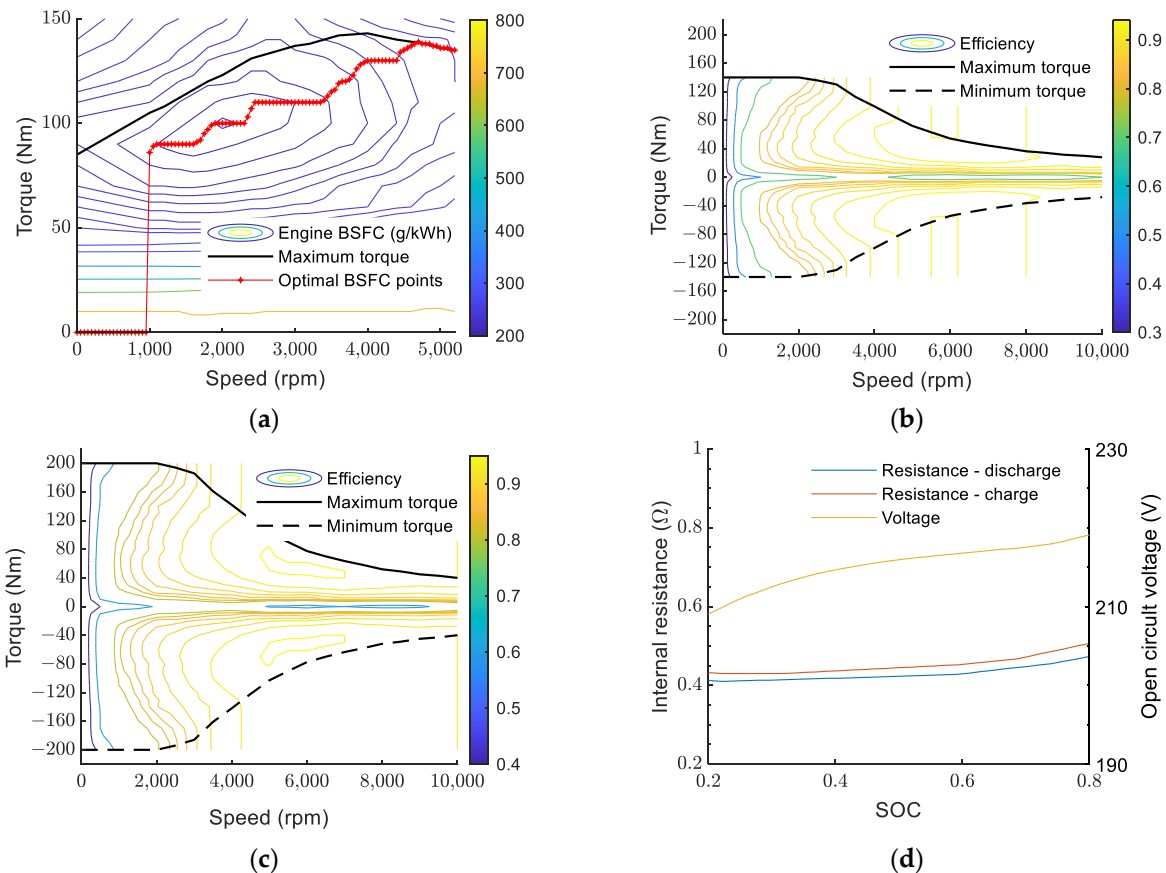

**Figure 2.** Component lookup tables for the powertrain: (**a**) engine, (**b**) MG1, (**c**) MG2, and (**d**) battery.

According to Figure 2a, the instantaneous fuel rate ($\dot{m}$) is described as a map of engine speed ($\omega_e$) and engine torque ($T_e$):

$$\dot{m} = f_1(\omega_e, T_e) \tag{6}$$

where $T_e$ is limited by the maximum value ($T_{e\_max}$), i.e., the black line in Figure 2a. Optimal brake-specific fuel consumption (BSFC) torque trajectory was calculated offline and is depicted by the dotted red line. Figure 2b,c show efficiency maps for both motors, and their consumed power was calculated as follows:

$$P_{mg1} = T_{mg1}\omega_{mg1}\eta_{mg1}^k(T_{mg1}, \omega_{mg1}) \tag{7}$$

$$P_{mg2} = T_{mg2}\omega_{mg2}\eta_{mg2}^k(T_{mg2}, \omega_{mg2}) \tag{8}$$

where $\eta_{mg1}^k$ and $\eta_{mg2}^k$ are operating efficiencies of MG1 and MG2, respectively, and the superscript $k$ signifies the mode of operation: $k$ equals 1 for motor operations and $-1$ for generator operations.

An equivalent circuit model was developed for the Li-ion battery system:

$$P_{end} = P_{mg1} + P_{mg2} = U_{oc}I_b - I_b^2 R_b \tag{9}$$

where $P_{end}$ is power at battery terminals, $I_b$ is current (positive during discharge, negative during charging), $U_{oc}$ is open-circuit voltage, and $R_b$ is equivalent internal resistance. As

shown in Figure 2d, both $U_{oc}$ and $R_b$ are related to battery state of charge (SOC). Through solving of Equation (9), $I_b$ can be determined as follows:

$$I_b = \frac{U_{oc} - \sqrt{U_{oc}^2 - 4P_{end}R_b}}{2R_b} \tag{10}$$

Therefore, the dynamic of SOC can be determined as follows:

$$\frac{dSOC}{dt} = -\frac{I_b}{Q_b} = -\frac{U_{oc} - \sqrt{U_{oc}^2 - 4P_{end}R_b}}{2R_bQ_b} \tag{11}$$

where $Q_b$ is battery capacity.

### 2.3. Hierarchical EC

With the help of vehicle-to-everything (V2X) technologies, it is possible to improve fuel efficiency of an HEV on two levels: energy management and speed control. There are two types of eco-cruising architectures: integrated and hierarchical. Even though the integrated type maximizes fuel savings, it suffers from a heavy computing burden as speed planning and energy management are solved simultaneously. As an alternative, we propose an EC based on hierarchical architecture, as illustrated in Figure 3.

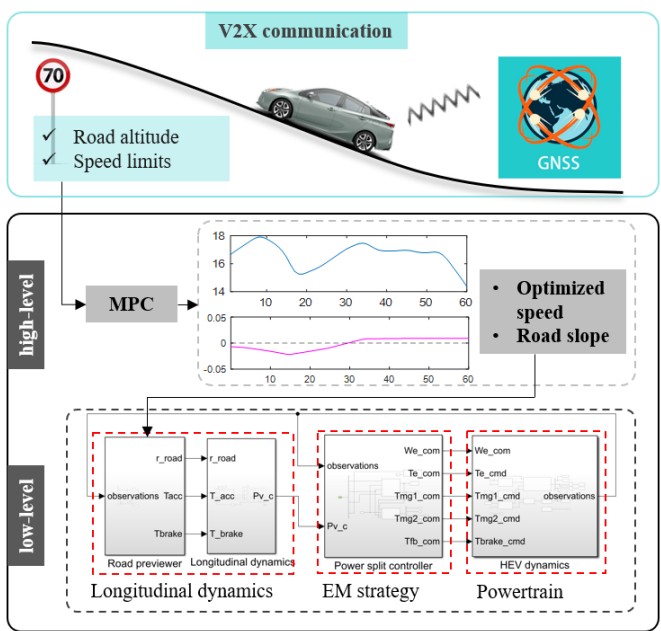

**Figure 3.** Hierarchical EC.

Using the information from V2X communication, the high-level controller is responsible for calculating optimal speed. The low-level powertrain controller then executes power allocation, based on optimized speed and road slope, in the powertrain. This work focuses only on the design of the high-level speed controller; the low-level powertrain controller directly adopts the heuristic strategy that is employed in actual vehicles.

## 3. MPC Method and RTI Algorithm

As shown in Figure 3, speed planning was the first step in the EC. In this section, an approximated fuel model is presented as a preliminary step towards developing MPC. Conventional MPC in time domain is then formulated, followed by our MPC in the space domain. Lastly, the RTI algorithm was proposed in order to accelerate the process of solving receding optimizations.

### 3.1. Approximated Fuel Model

In consideration of an equilibrium SOC, fuel is the only source of energy. Accordingly, it is reasonable to assume that all of a vehicle's demands are met by the engine, which is always operating on the BSFC (as shown in Figure 2a) [18].

Fuel consumption corresponding to different power demands is shown in Figure 4.

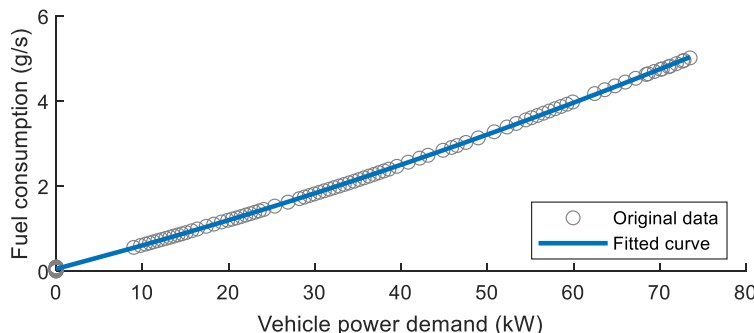

**Figure 4.** Fuel consumption corresponding to different power demands.

The solid blue line represents the fitted curve and is given by the following equation:

$$f_2(P_V) = \beta_2 P_V^2 + \beta_1 P_V + \beta_0 \tag{12}$$

where $P_V$ is power demand. It is evident that Equation (12) is a good match to the original data. Fitting parameters $\beta_2$, $\beta_1$, and $\beta_0$ equal $1.95 \times 10^{-10}$, $5.35 \times 10^{-5}$, and $4.96 \times 10^{-2}$, respectively.

### 3.2. MPC Formulation in Time Domain

Speed-planning MPC is conventionally formulated in the time domain, with a sampling interval of $\Delta t$. In this case, system states are vehicle speed ($v$) and vehicle location ($s$), while the control variable is acceleration ($a$). System state dynamics can be discretized as follows:

$$v^k(i+1) = v^k(i) + a(i)\Delta t \tag{13}$$

$$s^k(i+1) = s^k(i) + v^k(i)\Delta t + \frac{1}{2}a(i)\Delta t^2 \tag{14}$$

where $k$ is the instant when the prediction is made and $i$ is the discrete time index. The vehicle power demand is

$$
\begin{aligned}
P_V^k(i) \\
= v^k(i)\left[ ma(i) + mg\left(\sin\alpha\left(s^k(i)\right) + f\cos\alpha\left(s^k(i)\right)\right) + \tfrac{1}{2}C_D A_f \rho\left(v^k(i)\right)^2 \right] \\
= v^k(i)\left[ ma(i) + F_{road}\left(s^k(i)\right) + \tfrac{1}{2}k_{air}\left(v^k(i)\right)^2 \right]
\end{aligned}
\tag{15}
$$

where $F_{road}\left(s^k(i)\right)$ is road resistance and $k_{air}$ denotes $C_D A_f \rho$.

Speed-planning MPC in the time domain can be summarized as follows:

$$\min_{v^k,a} \sum_{i=1}^{N_t} f_2\left(P_V^k(i)\right)\Delta t \tag{16a}$$

*s.t.* Equations (13)–(15) *and*

$$v^k(i) \in \left[\underline{v}\left(s^k(i)\right), \bar{v}\left(s^k(i)\right)\right] \tag{16b}$$

$$a(i) \in [\underline{a}, \bar{a}] \tag{16c}$$

$$P_V^k(i) \in \left[\underline{P}_V, \bar{P}_V\right] \tag{16d}$$

$$v^k(1) = v^k, \quad s^k(1) = s^k \tag{16e}$$

$$v^k(N_t) \in \phi_f^v, \quad s^k(N_t) \in \phi_f^s \tag{16f}$$

where Equation (16a) gives the objective function, with $N_t$ representing the prediction horizon; Equations (16b–d) are constraints on system variables; and Equation (16e) sets the initial states to the current system states. In addition, final vehicle speed and vehicle location were specified by Equation (16f), where $\phi_f^v$ and $\phi_f^s$ are adjusted according to study scenario. We refer to MPC defined by Equation (16) as TMPC.

### 3.3. MPC Formulation in Space Domain

It is evident that in TMPC, both road resistance—$F_{road}\left(s^k(i)\right)$ and speed limits—$\underline{v}\left(s^k(i)\right)$, $\bar{v}\left(s^k(i)\right)$ depend on vehicle location $s^k(i)$. Therefore, $F_{road}\left(s^k(i)\right)$ has to be guessed in the time domain, which is an inherent drawback of the control in the time domain [24]. Solving the problem in the time domain would require fitting of $F_{road}\left(s^k(i)\right)$ as a nonlinear function of traveled distance. For long horizons and general slope profiles, this would be very difficult to carry out if we wanted to preserve accuracy. Another concern is speed limits. As we use them, speed limits are non-differentiable functions of distance, hence solving in the time domain would make the problem a mixed-integer nonlinear programming problem, which would be extremely difficult to solve.

To reduce computational complexity, we present a space-domain method with a sampling interval of $\Delta s$. When a road prediction is made, both road resistance and speed limits remain constant throughout MPC solving. Furthermore, speed limits can be handled very efficiently by most solvers, as they are simply box constraints.

The nominal kinetic ($E = v^2/2$) and its derivative with respect to distance are defined as:

$$\frac{dE}{ds} = v\frac{dv}{ds} = \frac{dv}{dt} = a \tag{17}$$

which is then discretized into:

$$E^k(i+1) = E^k(i) + a(i)\Delta s. \tag{18}$$

In the space-domain method, travel time ($t$) is now considered a state variable, and its discrete dynamics are given by:

$$t^k(i+1) = t^k(i) + \frac{\Delta s}{\sqrt{2E^k(i)}}. \tag{19}$$

In addition, vehicle power demand (Equation (15)) changes to:

$$P_V^k(i) = \sqrt{2E^k(i)}\left[ma(i) + F_{road}(i) + k_{air}E^k(i)\right]. \tag{20}$$

Speed-planning MPC in space domain can be summarized as follows:

$$\min_{E^k,a} \sum_{i=1}^{N_s} f_2\left(P_V^k(i)\right) \frac{\Delta s}{\sqrt{2E^k(i)}} \tag{21a}$$

*s.t.* Equations (18)–(20) *and*

$$E^k \in \frac{1}{2}\left[\underline{v}^2(i), \bar{v}^2(i)\right] \tag{21b}$$

$$a(i) \in [\underline{a}, \bar{a}] \tag{21c}$$

$$P_V^k(i) \in \left[\underline{P}_V, \bar{P}_V\right] \tag{21d}$$

$$E^k(1) = \frac{1}{2}\left(v^k\right)^2, \quad t^k(1) = k \tag{21e}$$

$$E^k(N_s) \in \phi_f^E, \quad t^k(N_s) \in \phi_f^t \tag{21f}$$

where Equation (21a) gives the objective function, with $N_s$ representing the prediction horizon; Equations (21b–d) are constraints on system variables; and Equation (21e) sets the initial states to the current system states. In addition, final vehicle speed and vehicle location are specified by Equation (21f), where $\phi_f^E$ and $\phi_f^t$ are adjusted according to study scenario. MPC defined by Equation (21) is referred to as SMPC. For ease of understanding, Figure 5 compares SMPC and TMPC.

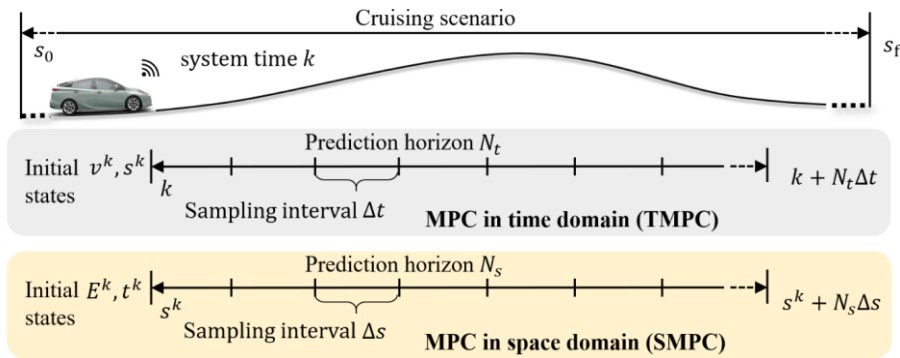

**Figure 5.** Comparison between TMPC and SMPC.

### 3.4. RTI Algorithm

The OCP in Equation (21) is nonlinear and non-convex, essentially making SMPC into non-convex NMPC. When the control system sets a short update interval, such as 0.1 s, computations must be completed in milliseconds. Unfortunately, NMPC is usually difficult to solve in real time, which limits its practical application in the automotive industry.

For NMPC to be more practicable, real-time optimization of controlled processes has become increasingly popular. Several real-time optimization methods are available, but RTI appears to be the most efficient, since it reduces computation time to a minimum while maintaining all the advantages of a fully nonlinear treatment [27]. Typical RTI focuses on convex NMPC and uses the sequential quadratic programming (SQP) algorithm, performing only one quadratic programming (QP) iteration per MPC update [26]. The RTI is seldom used in non-convex NMPC. We take a further step in this study and propose an RTI algorithm for non-convex NMPC (see Equation (21)).

Considering the sampling instants of the system—$[k_1, k_2, k_3, \ldots] = [0s, 0.1s, 0.2s, \ldots]$—the RTI algorithm would proceed as Algorithm 1:

---

**Algorithm 1** RTI

---

1: Start with the sampling instant $(k_1)$ and an initial guess for non-convex NMPC (16), i.e., $X_0^{k_1} = \left(E_0^{k_1}, t_0^{k_1}, a_0^{k_1}\right) = (\mathbf{0}, \mathbf{0}, \mathbf{0})$.

2: Perform the interior point (IP) algorithm until it converges, send optimal feedback control $\left(a_0^{k_1}(1)\right)$ to the system, and record the optimal solution as $X_{opt}^{k_1} = \left(E_{opt}^{k_1}, t_{opt}^{k_1}, a_{opt}^{k_1}\right)$. Move to the next sampling instant $(k_2)$.

3: Extract the acceleration sequence $\left(a_{opt}^{k_1}\right)$ from $X_{opt}^{k_1}$. Define $a_0^{k_2}$ as identical to $a_{opt}^{k_1}$, and generate $E_0^{k_2}$ and $t_0^{k_2}$ according to Equations (17) and (18). Construct the warm guess—$X_0^{k_2} = \left(E_0^{k_2}, t_0^{k_2}, a_0^{k_2}\right)$—that approximates exact optimal results.

4: Perform the IP algorithm with maximum iteration times $(N_{RTI})$. Send $a_0^{k_2}(1)$ as the optimal feedback control. Record the optimal solution as $X_{opt}^{k_2} = \left(E_{opt}^{k_2}, t_{opt}^{k_2}, a_{opt}^{k_2}\right)$.

5: Move to the next sampling interval and return to step (3).

---

Through adoption of small $N_{RTI}$ values, the RTI algorithm can significantly reduce computation time required by the IP. Extensive analysis will be conducted in Section 4 to evaluate performance of the proposed RTI algorithm.

## 4. Simulation and Results

### 4.1. Simulation Settings

#### 4.1.1. Reference Speed Controllers and EM Strategy

For comparison purposes, two reference speed controllers were developed: the constant-speed controller (CC) and the globally optimal controller (GOC). The GOC uses a custom DP algorithm involving efficient variable grids to achieve optimal speed throughout the entire driving journey [32].

As discussed in Section 2.3, a heuristic strategy was chosen for the low-level controller. In general, heuristic strategies use a vehicle's current states as input to determine operating mode and powertrain output [33]. Here, the vehicle states are engine on/off state (denoted by σ), vehicle power demand ($P_V$), battery (SOC) and vehicle speed ($v$). According to the published work in [34], we developed a heuristic strategy that is very similar to the real-life EM strategy. Table 1 provides a summary of operation modes as well as their decision rules. It should be noted that, in hybrid mode, desired battery power and engine output are determined by offline policies [34].

**Table 1.** Power-split HEV operation modes.

| Operation Modes | Decision Rules |
|---|---|
| Braking mode—mechanical braking | $P_V < 0 \wedge SOC \geq S\bar{O}C$ |
| Braking mode—regenerative braking | $P_V < 0 \wedge SOC < S\bar{O}C$ |
| Electric mode | $0 < P_V \leq \bar{P}_{ev} \wedge SOC \geq SOC_{targ} \wedge v \leq \bar{v}_{ev}$ |
| Hybrid mode | $\left\{ \sigma = 0 \wedge 0 < P_V \wedge SOC < SOC_{targ} \right\}$ $\vee \left\{ \sigma = 0 \wedge P_V > \bar{P}_{ev} \right\} \vee \left\{ \sigma = 0 \wedge v > \bar{v}_{ev} \right\}$ $\vee \left\{ \sigma = 1 \wedge SOC \leq SOC_{set} \right\}$ |

The thresholds used in the heuristic strategy are described in Table 2.

**Table 2.** Thresholds of the heuristic strategy.

| Thresholds | Decision Rules | Values |
|---|---|---|
| $S\bar{O}C$ | Highest battery SOC | 0.8 |
| $SOC_{targ}$ | Target battery SOC during discharge | 0.5 |
| $SOC_{set}$ | Battery SOC set point during charging | 0.65 |
| $\bar{P}_{ev}$ | Maximum power demand in electric mode | 9 kW |
| $\bar{v}_{ev}$ | Maximum vehicle speed in electric mode | 16 m/s |

A detailed explanation of the heuristic strategy can be found in [34] and is not repeated here. In order to calibrate the heuristic strategy, we compared the experimental and simulation results, and the experimental data is provided in [31]. The comparison is illustrated in Figure 6. It can be seen that the established heuristic strategy produced similar results to those obtained with vehicle testing. Vehicle testing showed 527.38 g of total fuel consumption while the simulation showed 520.23 g, representing a difference of only 1.36%.

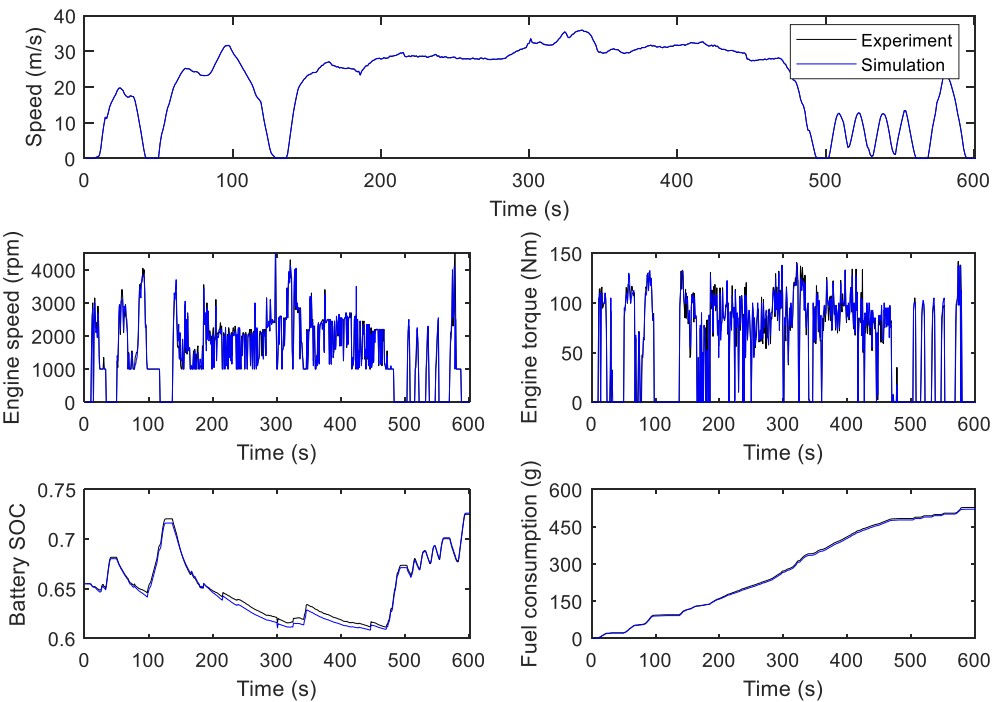

**Figure 6.** Heuristic strategy comparison between experiment and simulation.

It is important to note that the heuristic strategy does not possess optimality and could influence fairness of comparison between speed controllers. However, the heuristic strategy is widely used in reality, so it will provide the engineer with greater insight into application of speed controllers.

### 4.1.2. Simulation Parameters

Other HEV parameters used in this study can be found in Table 3.

**Table 3.** Power-split HEV Prius 2013 parameters.

| Parameters | Values |
| --- | --- |
| Vehicle mass $(m)$ | 1450 kg |
| Aerodynamic drag coefficient | 0.28 |
| Vehicle front area $\left(A_f\right)$ | 2.52 m$^2$ |
| Rolling resistance coefficient $(f)$ | 0.015 |
| Air density $(\rho)$ | 1.20 kg/m$^3$ |
| Gravitational acceleration $(g)$ | 9.81 m/s$^2$ |
| Wheel radius $(r)$ | 0.28 m |
| Final reduction ratio $\left(A_f\right)$ | 3.30 |
| PGSs parameter $(R_1/S_1,\ R_2/S_2)$ | 3.60, 2.63 |
| Maximum engine power/torque | 73 kW/142 Nm |
| Fuel consumed in restarting of engine | 0.60 g |
| MG1 maximum power/torque | 42 kW/140 Nm |
| MG2 maximum power/torque | 60 kW/200 Nm |
| Battery capacity $(Q_b)$ | 1.35 kWh |

The following simulation was performed on a desktop computer (Intel i7-10700 CPU at 2.9 GHz and 16 GB RAM), using Matlab 2021b. Additionally, CasADi software with the IPOPT solver was selected to perform IP iteration [35].

*4.2. Results and Discussion*

4.2.1. Speed Planning Results

We conducted the EC on two real-life roads featuring many hills: the urban expressway between Xuanwu District Nanjing and Jurong Zhenjiang, and the motorway between Chengdu and Chongqing. Figure 7 presents road altitude derived from Google Elevation API. The high speed limit on the expressway is 80 km/h as required by law, while the low speed limit was artificially set at 60 km/h. In the case of the motorway, both the high and low speed limits were set at 100 km/h and 60 km/h, respectively.

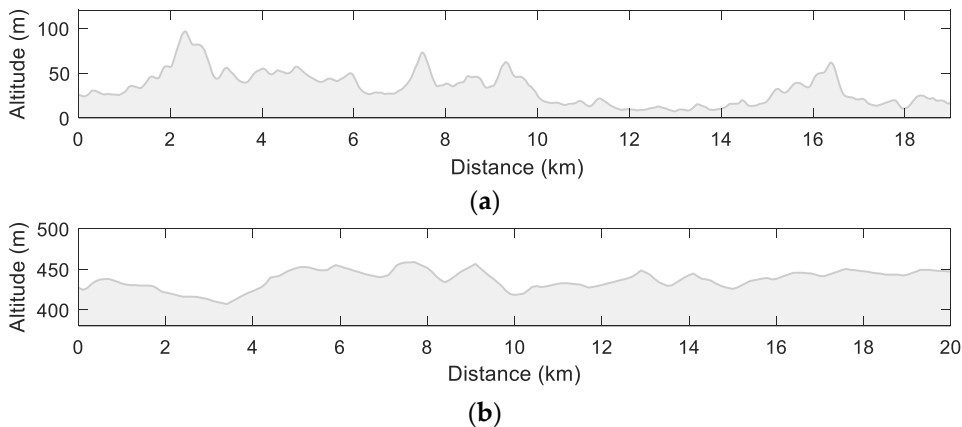

**Figure 7.** Two testing roads: (**a**) the expressway between Nanjing and Zhenjiang (19 km), and (**b**) the motorway between Chengdu and Chongqing (20 km).

The prediction distance was set at a default of 1000 m. Using an average speed of 70 km/h for the expressway, the desired travel time was 51.43 s. Furthermore, we hoped that the HEV would complete the predicted distance at a speed of 70 km/h. As a result, $\phi_f^v$, $\phi_f^s$, $\phi_f^E$, and $\phi_f^t$ in TMPC and SMPC were set to 19.44 m/s, 1000 m, 189.03, and 51.43 s, respectively. The prediction horizons, $N_t$ and $N_s$, were each set to 50, while the sampling intervals, $\Delta t$ and $\Delta s$, were set to 1.0286 s and 20 m, respectively.

In the case of the motorway, the average speed was clearly set at 80 km/h. Considering the prediction horizon of 1000 m, the desired travel time was 45 s. We could easily obtain the MPC design parameters $\phi_f^v$, $\phi_f^s$, $\phi_f^E$, $\phi_f^t$, $N_t$, $N_s$, $\Delta t$, and $\Delta s$, as 22.22 km/h, 1000 m, 246.91, 45 s, 50, 50, 0.9 s, and 20 m, respectively. In the RTI algorithm, the maximum iteration times ($N_{RTI}$) were set as 8.

For TMPC to impose road resistance, the nonlinear function $F_{road}(s)$ is required. In order to fit actual road resistance, we selected the sum of sines, which is expressed as follows:

$$F_{road}(s) = \sum_{j=1}^{8} b_j \sin(c_j s + d_j) \tag{22}$$

where $b_j$, $c_j$, and $d_j$ are fitting parameters that need to be updated periodically. As an example, Figure 8 shows road resistance along the first predicted distance. It is evident that Equation (22) offers a reasonable approximation for the actual data.

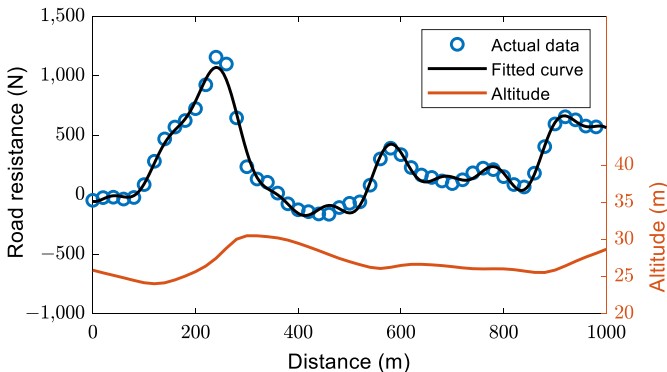

**Figure 8.** Road resistance along the distance.

Road altitude and vehicle speed results on the expressway are demonstrated in Figure 9. Because the EC system updates every 0.1 s, all of the following results are described in time coordinates. As can be seen, all speeds were strictly constrained within limits. By correlating SMPC speed results with altitude in Figure 9a, we can derive the following speed-planning features: acceleration before going uphill, deceleration during the uphill ascent, and acceleration during the downhill descent.

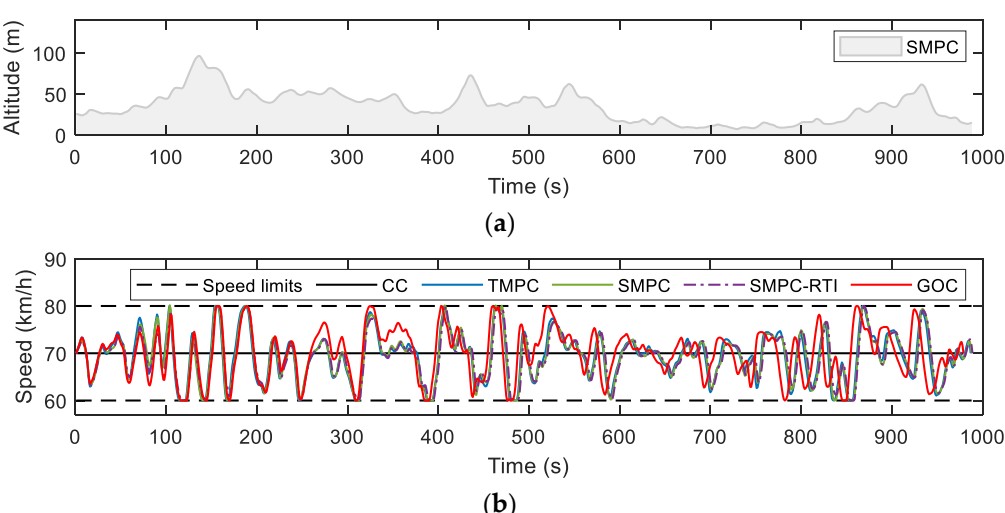

(**a**)

(**b**)

**Figure 9.** Road altitude and vehicle speed results on the expressway. (**a**) Road altitude against driving time in SMPC. (**b**) Vehicle speed results of different speed controllers.

Compared to the MPC results, the speed trajectory of the GOC clearly shows a different amplitude. This is because the GOC forecasts the entire road, and DP is able to find possible optimal solutions. In addition, the GOC leads to a shorter travel time than that of other speed controllers.

In Figure 9b, it is evident that the three MPCs exhibited quite similar speed trajectories. A zoomed-in comparison of the three MPCs is illustrated in Figure 10. It is evident that the SMPC-RTI exhibits close results to those of SMPC, which indicates that the proposed RTI maintains good solution quality.

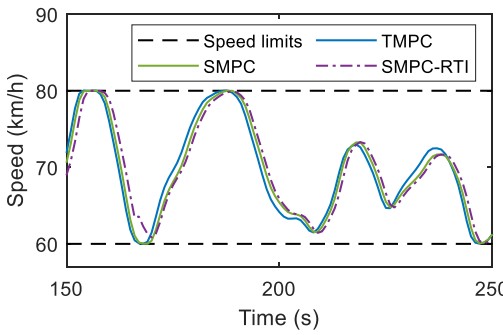

**Figure 10.** Comparison between three MPCs.

Figure 11 shows acceleration of all speed controllers. It can be seen from Figure 11 that vehicle acceleration was controlled within $[-1, 1]$ m/s$^2$, which ensured acceptable driving comfort for passengers.

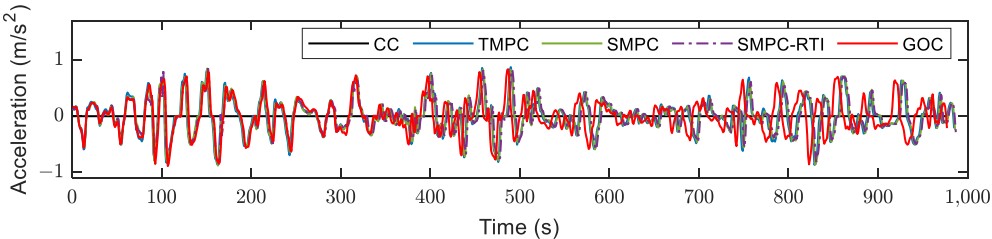

**Figure 11.** Acceleration of different speed controllers on the expressway.

Road altitude and results on the motorway are demonstrated in Figure 12. As can be seen, all MPC controllers and GOCs exhibited similar speed and acceleration results. Since the terrain in Figure 12a contains fewer hills than the terrain depicted in Figure 9a, the speed regulation on the motorway appears smoother than that on the expressway. Moreover, although there was a wide range of flexibility available, the controllers only regulated speed between [69, 91] km/h. This indicates that maximizing speed band usage may not always be necessary for optimal performance.

### 4.2.2. Powertrain Results

This subsection presents powertrain results, obtained with the EC, that incorporated the high-level speed controller and the low-level powertrain controller. As a convenience, we have included only the expressway case in this subsection.

Figure 13 illustrates SOC and battery power. As the heuristic strategy cannot guarantee SOC equality strictly, initial SOC in each EC was manually determined. Table 4 lists initial and terminal SOC of each controller; the two values are close to each other.

It is shown in Figure 13a that, when compared to the EC that adopted the CC, the ECs that adopted other speed controllers revealed a different SOC pattern. Similar phenomena can be observed in Figure 13b. Meanwhile, when speed was optimized, battery power trajectory appeared smoother.

An illustration of the engine points distribution is shown in Figure 14, along with a statistic regarding engine speed and torque.

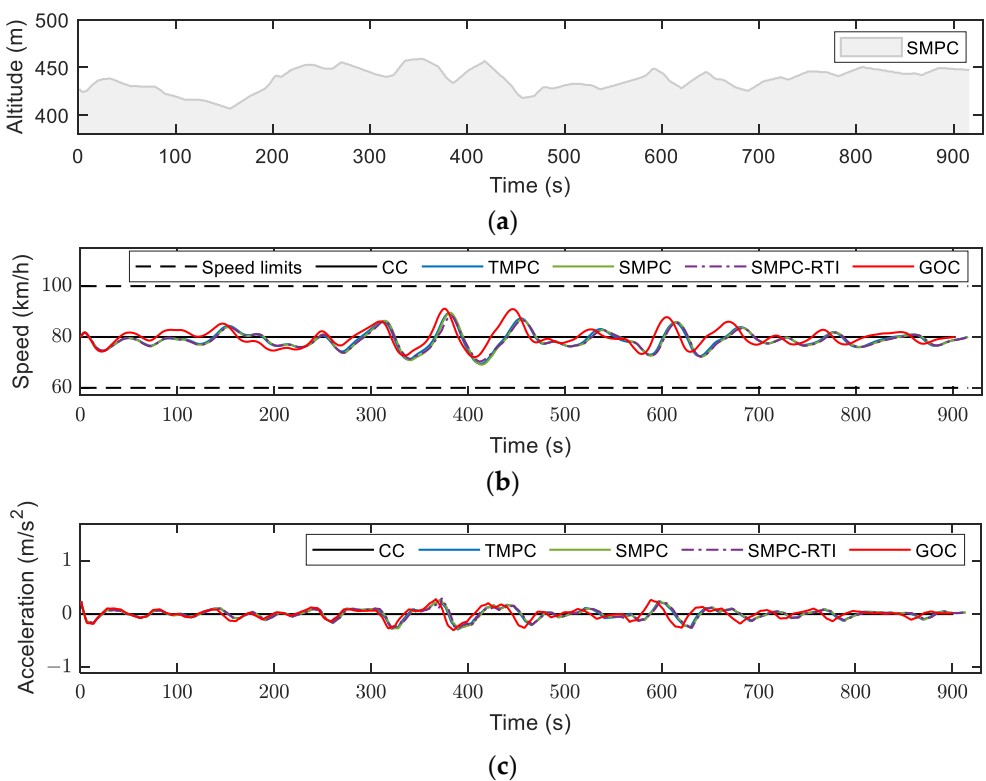

**Figure 12.** Road altitude and results on the motorway. (**a**) Road altitude against driving time in SMPC. (**b**) Vehicle speed results of different speed controllers. (**c**) Acceleration of different speed controllers.

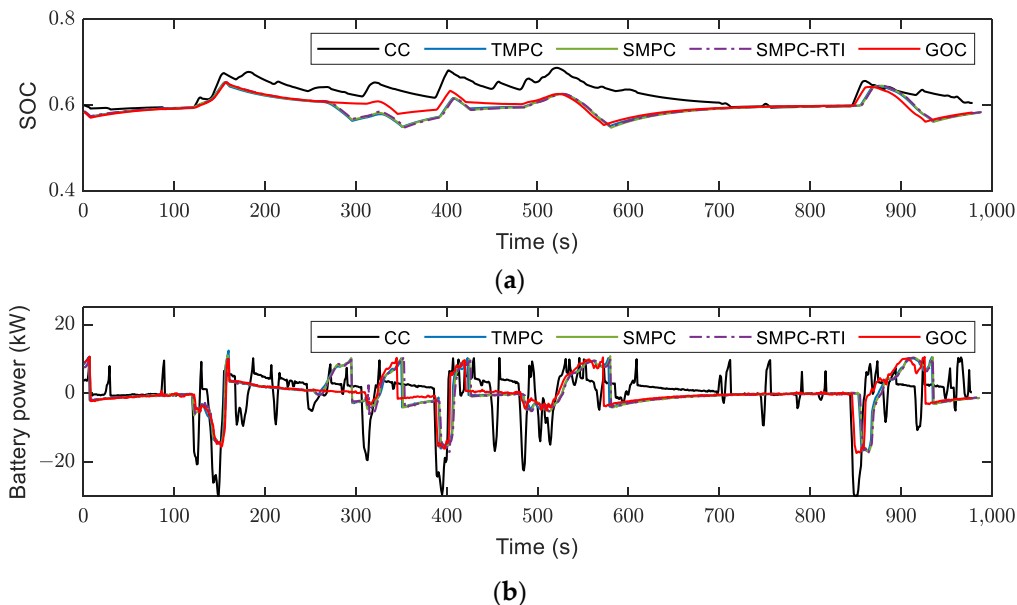

**Figure 13.** SOC and battery power of the EC using different speed controllers: (**a**) SOC, (**b**) battery power.

**Table 4.** Initial and terminal SOC of the EC using different speed controllers.

|  | CC | TMPC | SMPC | SMPC-RTI | GOC |
|---|---|---|---|---|---|
| Initial SOC | 0.6000 | 0.5850 | 0.5850 | 0.5870 | 0.5850 |
| Terminal SOC | 0.6030 | 0.5832 | 0.5829 | 0.5854 | 0.5832 |

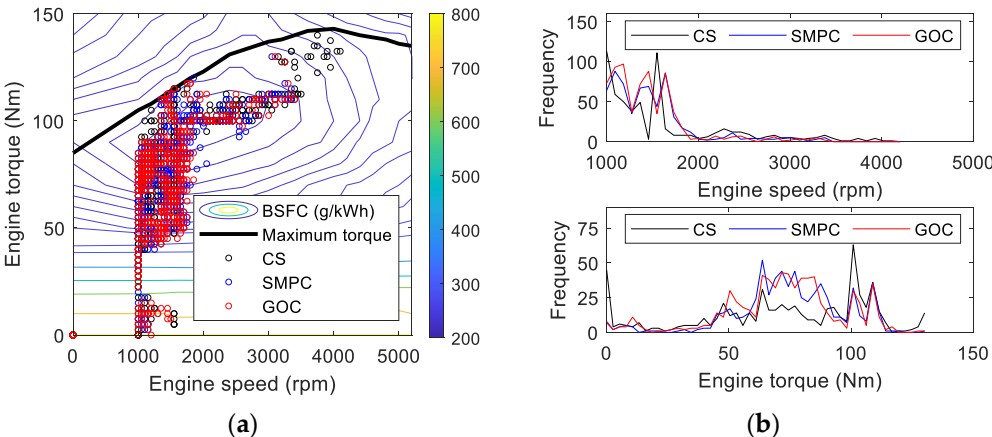

**Figure 14.** Engine results for the EC using different speed controllers: (**a**) engine-point distribution and (**b**) histogram of engine speed and torque.

It is clear from Figure 14b that through incorporation of SMPC and the GOC, the EC allows more engine operation at low speeds ([1100, 1900] rpm) and medium torque ([50, 90] Nm), thereby increasing fuel efficiency.

### 4.2.3. Fuel-Consumption Evaluation

Table 5 presents performance of the EC using different speed controllers on the expressway. As shown in Table 5, average speed in each group remained relatively close, as did SOC deviation. This contributed to fairness of the next fuel comparison. Using the CC, the EC consumed 558.22 g of fuel. Fuel consumption decreased significantly after other speed controllers were introduced. As compared to the CC, the GOC helped the EC save 8.06% of fuel. TMPC, SMPC, and SMPC-RTI also achieved fuel savings of 7.84%, 7.90%, and 7.68% for the EC, respectively. In conclusion, the proposed TMPC and SMPC provide suboptimal solutions to the speed-planning problem. Furthermore, the proposed RTI algorithm preserves quality of solution for NMPC.

**Table 5.** Performance of the EC using different speed controllers.

|  | CC | TMPC | SMPC | SMPC-RTI | GOC |
|---|---|---|---|---|---|
| Average speed (km/h) | 70 | 69.67 | 69.63 | 69.54 | 69.85 |
| SOC deviation | 0.0030 | −0.0018 | −0.0021 | −0.0016 | −0.0018 |
| Fuel consumption (g) | 558.22 | 514.48 | 514.12 | 515.31 | 513.24 |
| Fuel savings (%) | - | 7.84 | 7.90 | 7.68 | 8.06 |

Table 6 presents performance of the EC using different speed controllers on the motorway. In this case, the GOC assisted the EC in reducing fuel consumption by 4.30% compared to the CC. Furthermore, TMPC, SMPC, and SMPC-RTI achieved fuel savings of 4.32%, 4.30%, and 4.50% for the EC, respectively.

**Table 6.** Performance of the EC using different speed controllers.

|  | CC | TMPC | SMPC | SMPC-RTI | GOC |
|---|---|---|---|---|---|
| Average speed (km/h) | 80 | 79.27 | 79.37 | 79.37 | 79.82 |
| SOC difference * | −0.0006 | −0.0007 | −0.0008 | −0.0007 | −0.0008 |
| Fuel consumption (g) | 632.17 | 605.01 | 604.84 | 604.96 | 603.72 |
| Fuel savings (%) | - | 4.30 | 4.32 | 4.30 | 4.50 |

* SOC difference is deviation between initial and terminal SOC in Table 4.

Please note that the fuel savings above are only attributed to the high-level speed controller. A further reduction in fuel consumption could be achieved via incorporation of advanced EM strategies into the low-level powertrain controller.

### 4.2.4. Computation-Time Evaluation

Figure 15 illustrates computation-time statistics for all MPC-based speed controllers. The mean computation time of the TMPC was 27.68 ms, which indicates that the optimal control for the current system was not available after 27.68 ms. In other words, the EC needs to perform the previous optimal control within 27.68 ms, which for the current system state was not optimal. Note that the sampling interval is 100 ms; therefore, the mean computation time of 27.68 ms may have negatively affected MPC performance in real time [27]. Furthermore, the fitting process of Equation (22) may be time-consuming, which would also impede application of conventional TMPC.

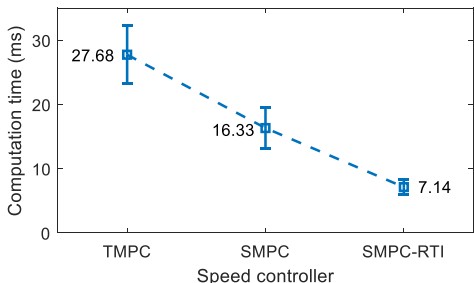

**Figure 15.** Computation-time statistics for speed controllers.

As shown in Figure 15, the mean computation time of the proposed SMPC was 16.33 ms, which represents a 41% reduction over that of TMPC. Through use of RTI, the mean computation time was further reduced to 7.14 ms. It was possible to accelerate that 7.14 ms by adopting a smaller $N_{RTI}$. Consequently, we performed a sensitivity analysis of $N_{RTI}$; results are shown in Figure 16.

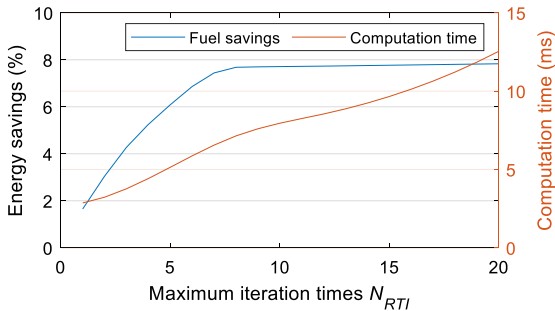

**Figure 16.** Results of the sensitivity analysis.

As shown in Figure 16, energy savings increase very slowly after $N_{RTI}$ equals 8. Consideration of the two indicators together suggests that $N_{RTI}$ should be set at 8.

### 5. Real-Time Implementation

An HIL test platform was established, as shown in Figure 17a, in order to examine the proposed SMPC's real-time performance. In Figure 17b, the schematic for the HIL test is shown; the HEV model and the proposed EC were run on the dSPACE® MicroLabBox and the dSPACE® MicroAutoBox, respectively. CAN cables were used to connect the two testing boxes. Since MicroAutoBox has close hardware specifications to the onboard ECU, its test results are highly indicative of real-world behavior [36].

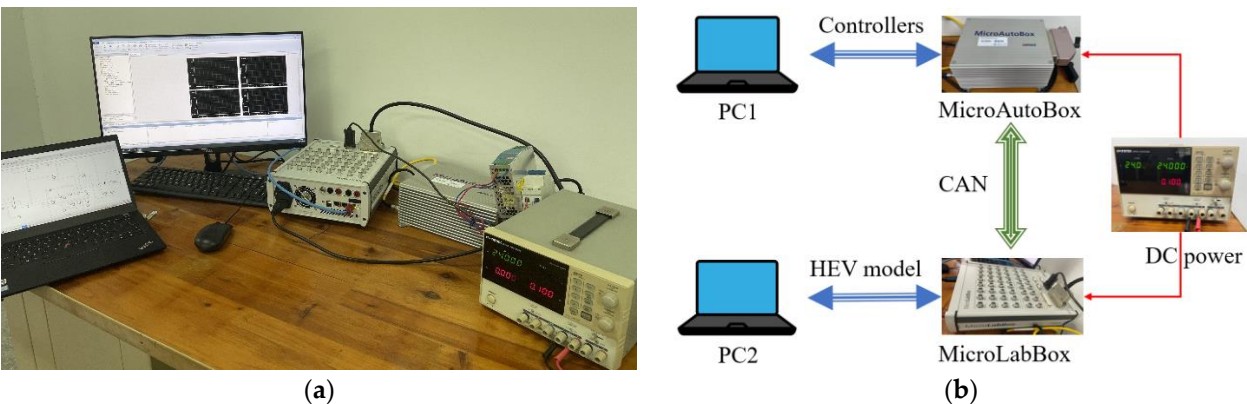

**Figure 17.** HIL (**a**) test platform and (**b**) schematic.

On the HIL test platform, ECs that included three different speed controllers—the CC, SMPC and SMPC-RTI—were evaluated. Real-time performance of the EC that used SMPC and SMPC-RTI is summarized in Table 7.

**Table 7.** Real-time performance of ECs using two speed controllers: SMPC and SMPC-RTI.

|  | SMPC | SMPC-RTI |
| --- | --- | --- |
| Fuel savings (%) | 7.88 | 7.67 |
| Mean computation time (ms) | 23.65 | 9.86 |

Table 7 illustrates that fuel savings achieved by HIL testing are comparable to those achieved via simulation in Table 5. The reason for this is that the same HEV model and sampling interval were adopted. In terms of computation time, both SMPC and SMPC-RTI displayed longer values in HIL tests than in simulations. According to Table 7, the mean computation time for SMPC-RTI was 9.86 ms, which is less than 1/8 of the sampling interval. SMPC-RTI is therefore capable of meeting requirements of potential onboard applications.

## 6. Conclusions

In this paper, we propose a real-time NMPC for a connected power-split HEV that is capable of optimizing driving speed on roads that feature many hills. Our proposed SMPC does not rely on guesses regarding road resistance, since it was designed in the space domain. When it was used as the top-level speed controller in a hierarchical EC, 7.90% and 4.32% fuel savings were achieved on a realistic expressway and on a motorway; this is slightly lower than the results for the EC that used the GOC. In addition, SMPC reduced computation time by 41% when compared to conventional TMPC that is designed in the time domain.

For the purpose of enhancing real-time performance, an RTI algorithm for solving SMPC, in which the solution of each previous update is used to generate warm initialization, was developed. Simulation results demonstrated that SMPC-RTI not only provides sufficient solution quality but also significantly reduces computation time. Testing on a HIL platform demonstrated that SMPC-RTI achieved a mean computation time of 9.86 ms, which suggests that it may be suitable for advanced real-time cruise control systems.

Future research should focus on modeling uncertain speed limits caused by traffic flow, which will enhance robustness of speed controllers.

**Author Contributions:** Conceptualization, F.J. and Q.W.; methodology, F.J.; software, F.J. and Y.Z.; validation, F.J. and W.Z.; formal analysis, F.J. and Y.Z.; investigation, F.J. and L.W.; resources, L.W.; data curation, F.J. and Y.Z.; writing—original draft preparation, F.J.; writing—review and editing, F.J. and Y.Z.; visualization, F.J.; supervision, L.W.; project administration, W.Z. and L.W.; funding acquisition, L.W. All authors have read and agreed to the published version of this manuscript.

**Funding:** This research was supported in part by the National Natural Science Foundation of China (Grant No. 52172383).

**Institutional Review Board Statement:** Not applicable.

**Informed Consent Statement:** Not applicable.

**Data Availability Statement:** Not applicable.

**Conflicts of Interest:** The authors declare no conflict of interest.

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
