# Peer review of "Real-Time NMPC for Speed Planning of Connected Hybrid Electric Vehicles"

_machines, doi:10.3390/machines10121129_

Round 1

Reviewer 1 Report

The paper proposes a method to improve computational time of an algorithm for eco-cruising of a power-split Hybrid Electric Vehicle (HEV). More in deep, authors present a time-domain MPC (TMPC) and a space-domain MPC (SMPC) to leverage information on altitude and plan the vehicle velocity accordingly. Then, a Real Time Iteration (RTI) algorithm is implemented with the SMPC to further reduce computational time and its real-time capabilities are tested on a Hardware-in-the-Loop (HIL) environment.

The reviewer would positively comment on the methodology section and the soundness of it. However, contributions of the paper may not be exhaustive for a journal article considering that SMPC have been often implemented for eco-driving leaving the RTI implementation as the only contribution. Authors might consider this type of analysis for a conference proceeding.

A detailed transcript of needed improvements are listed below for the authors’ convenience.

·        The reviewer would suggest to include more routes, as it seems that the one considered is a fairly simple one where minimum and maximum speed limits are very close and not a lot of scenarios are possible.

·        Not so clear from Figures 7 to 10 if the x-axis should be distance instead of time as it is in Fig. 6.

·        How did the authors calibrate the heuristic energy management strategy of the HEV? The values in Table 2 do not have a logic explained and the reviewer would suggest to include it.

·        Heuristic controls lack optimality of the solution, possibly not ensuring the fairness of the comparison between the velocity optimizers. The reviewer would suggest to include this limitation in the text, possibly in the introduction.

·        It is not clear to the reviewer what “may result in poor real-time performance” mean, in line 361. What do the authors mean with “poor”? Even though the computational time of the TMPC is higher, would it still be possible to implement it real-time?

Author Response

1. The reviewer would suggest to include more routes, as it seems that the one considered is a fairly simple one where minimum and maximum speed limits are very close and not a lot of scenarios are possible.

Response: Thanks for this comment. To test the speed controllers more thoroughly, we have included another motorway between Chengdu and Chongqing. Both the high and low speed limits are set at 100 km/h and 60 km/h, respectively. The detailed terrain can be found in Figure 7(b).

Road altitude and results on the motorway are demonstrated in Figure 12. As can be seen, all MPC and GOC controllers exhibit similar speed and acceleration results. Since the terrain in Figure 12(a) contains less saddles than the terrain depicted in Figure 9(a), the speed regulation on the motorway appears smoother than that on the expressway. More-over, although there is a wide range of flexibility available, the controllers only regulate the speed between [69,91] km/h. This indicates that maximizing the speed band usage may not always be necessary for optimal performance.

Table 6 presents the performance of the EC using different speed controllers on the motorway. In this case, the GOC assists the EC in reducing fuel consumption by 4.50% compared to the CC. Furthermore, the TMPC, SMPC, and SMPC-RTI achieve fuel savings of 4.30%, 4.32%, and 4.30% for the EC, respectively.

2. Not so clear from Figures 7 to 10 if the x-axis should be distance instead of time as it is in Fig. 6.

Response: Thanks for this comment. Because the EC system updates every 0.1 s, all the following results have been described in time coordinates. To clarify the terrain of the road, Figure 7 is added, in which the altitude is described in distance coordinates.

3. How did the authors calibrate the heuristic energy management strategy of the HEV? The values in Table 2 do not have a logic explained and the reviewer would suggest to include it.

Response: Thanks for the valuable comments. The detailed explanation about the heuristic strategy can be found in [36] and not repeated in this paper. In order to calibrate the heuristic strategy, we compare the experimental and simulation results, and the experimental data is provided in reference [31]. The comparison is illustrated in Figure 6. It can be seen that the established heuristic strategy produces similar results to those obtained with the vehicle testing. The vehicle testing shows 527.38 g of total fuel consumption while the simulation shows 520.23 g, which represents a difference of only 1.36%.

4. Heuristic controls lack optimality of the solution, possibly not ensuring the fairness of the comparison between the velocity optimizers. The reviewer would suggest to include this limitation in the text, possibly in the introduction.

Response: Thank you for your kind notification. In Section 4.1.1, we have discussed the limitations of adopting heuristic strategy as an EM strategy. Here are the revisions:

It is important to note that the heuristic strategy does not possess optimality and could influence the fairness of the comparison between speed controllers. However, the heuristic strategy is widely used in reality, so it provides the engineer with greater insight into the application of speed controllers.

5. It is not clear to the reviewer what “may result in poor real-time performance” mean, in line 361. What do the authors mean with “poor”? Even though the computational time of the TMPC is higher, would it still be possible to implement it real-time?

Response: Thanks for this comment. In principle, the optimal control associated with the current state of the system at time k will only become available at time k+27.68ms, once the computation has been completed. As a result, the EC needs to perform the previous optimal control within 27.68 ms, which is not optimal for the current state. Note that the sampling interval is 100 ms, therefore the mean computation time of 27.68 ms may negatively affect the real-time performance of the MPC. An in-depth discussion of practical real-time optimal control can be found in [27].

When the sampling interval is long enough, such as 1 s, it is possible to implement the TMPC in real-time. As our goal is to provide an efficient solution for applications that require short sampling intervals, the SMPC outperforms the TMPC exactly.

Reviewer 2 Report

This papers presents a valuable solution to the real-time implementation of eco-cruising controllers. The methods are  sound and the results, although in the simulation world,  show a promising advance in  terms of energy consumption, execution time.

Author Response

Thanks for the comments.

Reviewer 3 Report

-paper is written in a fair English, with a few typos or other small errors easy to correct with a last proofreading

-organization is clear and the sequence is easy to follow for the reader

-the theme is up-to-date and the modelling and control of HEV is interesting

-the contributions of using a space domain MPC and RTI are fair, even though the novelty is reduced

-simlations and HIL tests are interesting, even thogh only one case is presented

further details:

-missing units

-lines 22-24: redundant

-fig.2 what is the interest in plotting numbers ones on top of the others

-fig.3 increase font size in low-level ports for better legibility; what is the ouput of the high level ?

-fig.7 : we hardly can tel where are the variants CC/TMPC/SMPC only GOC is easily identified: even if the idea is to show they have similar results one wonders if something is missing!

-fig.8 there is a delay between SMPC and SMPC-RTI : the conclusion of lines 313,314 do not seem sustained ! 

-comparison of fig.10 is not obvious: can’t you find a metric to benchmark the battery usage ?

-again, the interpretation of fig.11b is not so obvious: can’t you define a metric to compare?

-table 4: is it the average or final SOC? the value is very small !?

Author Response

1. missing units

Response: Thank you for your notification. The manuscript has been carefully reviewed, and the missing units have been provided.

2. lines 22-24: redundant

Response: Thank you for your notification. Lines 22-24 have been rewritten to demonstrate the effectiveness of the RTI algorithm.

3. fig.2 what is the interest in plotting numbers ones on top of the others

Response: Thanks for this comment. To improve the clarity of the contour lines, we have adopted color labels and revised Figure 2.

4. fig.3 increase font size in low-level ports for better legibility; what is the ouput of the high level?

Response: Thanks for this comment. In low-level ports, the font size has been increased. As demonstrated in Figure 3, the output of the high level is the optimal speed and road slope.

5. fig.7: we hardly can tell where are the variants CC/TMPC/SMPC only GOC is easily identified: even if the idea is to show they have similar results one wonders if something is missing!

Response: Thanks for the valuable comments. The TMPC refers to the optimal control problem (OCP) (16), the SMPC and the GOC refer to the OCP (21). The OCP (16) is equivalent to the OCP (20) in principle. Therefore, it is understandable that different controllers produce similar results.

6. fig.8 there is a delay between SMPC and SMPC-RTI: the conclusion of lines 313,314 do not seem sustained!

Response: Thanks for the valuable comments. We have revised the conclusion of lines 339-340 to indicate that the proposed RTI offers good quality solutions.

7. comparison of fig.10 is not obvious: can’t you find a metric to benchmark the battery usage?

Response: Thanks for the valuable comments. Table 4 lists the initial and terminal SOCs of each controller, where the two values are close to each other.

8. again, the interpretation of fig.11b is not so obvious: can’t you define a metric to compare?

Response: Thanks for the valuable comments. Figure 14b presents the histogram of engine speed and torque. It is clear from Figure 14b that by incorporating the SMPC and GOC, the EC allows more engine operation at low speeds ([1100,1900] rpm) and medium torque ([50,90] Nm), thereby increasing fuel efficiency.

9. table 4: is it the average or final SOC? the value is very small?

Response: Thanks for this comment. The SOC difference is the deviation between the initial and terminal SOCs in Table 6.

Round 2

Reviewer 1 Report

The reviewer would still suggest to use different cycles, the two proposed are to easy to be run, no stops or traffic lights, no real scenarios are there. However, that could be for further development.